# Knocking Out Rap1a Attenuates Cardiac Remodeling and Fibrosis in a Male Murine Model of Angiotensin II-Induced Hypertension

**DOI:** 10.3390/cells14221834

**Published:** 2025-11-20

**Authors:** Cody S. Porter, Larissa T. Brown, Can’Torrius Lacey, Mason T. Hickel, James A. Stewart

**Affiliations:** Department of BioMolecular Sciences, School of Pharmacy, University of Mississippi, University, MS 38677-1848, USA

**Keywords:** angiotensin II (AngII), cardiac remodeling, hypertension, receptor for advanced glycation end-products (RAGE), repressor/activator protein 1a, RAS-related protein 1A (Rap1a)

## Abstract

Hypertension is a leading risk factor for cardiovascular disease and is associated with maladaptive cardiac remodeling, including hypertrophy and fibrosis. The roles of the receptor for advanced glycation end-products (RAGE) and the small GTPase Rap1a in angiotensin II (AngII)-induced remodeling remain unclear. This study examined how RAGE and Rap1a influence cardiac responses to AngII using wild-type (WT), RAGE knockout (RAGE KO), and Rap1a knockout (RapKO) mice. Cardiac structure and function were evaluated following AngII infusion. RapKO mice were protected from AngII-induced hypertrophy, whereas RAGE KO mice exhibited altered remodeling patterns. AngII consistently increased left ventricular wall thickness across all genotypes, indicating that structural remodeling is primarily treatment-driven. Measures of cardiac output and stroke volume also changed significantly with AngII, suggesting hemodynamic load as a key driver of functional adaptation. In contrast, diastolic functional parameters were genotype-dependent and remained stable with AngII exposure, demonstrating an intrinsic influence of RAGE and Rap1a on myocardial relaxation. These findings highlight distinct roles for RAGE and Rap1a in modulating hypertensive cardiac remodeling and may parallel human hypertensive heart disease, where increased RAGE and Rap1a expression associate with fibrosis and impaired relaxation. Targeting the crosstalk between the RAGE-AT1R axis and the cAMP-EPAC-Rap1a pathway may offer therapeutic potential to reduce adverse cardiac remodeling in hypertension.

## 1. Introduction

Hypertension is a primary risk factor for cardiovascular disease, with maladaptive cardiac remodeling representing a consequence of chronic hypertensive states [1,2,3]. RAGE (receptor for advanced glycation end-products), Rap1a (RAS-related protein 1A), and AngII (angiotensin II) form an interconnected signaling network that promotes to cardiac hypertrophy, inflammation, and fibrosis [4,5,6]. AngII, an effector of the renin–angiotensin–aldosterone system (RAAS), drives cardiomyocyte growth and extracellular remodeling through the AngII type 1 receptor (AT_1_R), stimulating downstream pathways such as mitogen-activated protein kinases (MAPKs) and phosphoinositide 3-kinase (PI3K)/Akt [7,8]. Evidence indicates reciprocal crosstalk between RAGE and AT_1_R, wherein activation of one receptor enhances expression of the other, establishing a feed-forward loop that amplifies oxidative stress and inflammation [8,9]. Inhibition of either receptor attenuates pathological remodeling, suggesting that targeting the RAGE–AT_1_R feedforward axis may represent a promising therapeutic strategy [10,11,12].

Rap1a, a small GTPase, regulates vascular tone and permeability through integrin-mediated signaling and endothelial barrier function, as well as fibroblast activity, and inflammatory signaling [13,14,15,16,17,18]. Additionally, dysregulation of Rap1a has been linked to endothelial dysfunction, a hallmark of hypertension [19,20,21]. Previous studies have demonstrated that AngII modulates inflammatory cytokine expression through Rap1a-dependent mechanisms, with Epac1 (exchange protein directly activated by cAMP 1) serving as a key intermediary [22,23]. AngII treatment has also been shown to decrease Rap1 activity in a dose-dependent manner while modulating sodium/hydrogen exchanger (NHE3) expression via cAMP-regulated pathways [22]. Furthermore, Rap1a and Epac1 signaling has been implicated in AngII-induced inflammatory cytokine production (IL-1β, IL-6, IL-8, TNF-α), underscoring its role in cardiovascular pathology [22,23]. While Rap1a and Epac1 signaling exhibits both maladaptive and protective effects in cardiac remodeling, Epac1 has been shown to promote hypertrophic remodeling by enhancing extracellular matrix deposition and fibrosis, whereas Epac2 has been associated with cardioprotective effects [13,16,24,25,26,27,28,29,30]. The differential roles of Epac isoforms highlight the complexity of cAMP-dependent Rap1a signaling in cardiovascular function and the need for further research.

We hypothesize that RAGE and Rap1a play critical roles in modulating the hypertensive response by influencing AngII-induced cardiac remodeling. This study investigated the effects of AngII treatment on cardiac parameters in wild-type (WT), RAGE knockout (RAGE KO), and Rap1a knockout (RapKO) mice. Our results indicate that while AngII significantly increased ventricular weight (VW) and the ventricular weight-to-body weight (VW/BW) ratio in WT and RAGE KO mice, RapKO mice exhibited resistance to these hypertrophic effects. Additionally, echocardiographic analyses revealed treatment-driven structural remodeling, most notably in left ventricular wall thickness with limited genotype-specific effects on systolic function. AngII treatment significantly increased myocardial collagen deposition in WT mice, whereas both RAGE and Rap1a deficiency markedly attenuated this fibrotic response. The protection observed in RapKO mice suggested that Rap1a contributes to maladaptive remodeling and fibrosis, supporting the potential of targeting Rap1a inhibition as a novel strategy to mitigate hypertensive cardiac pathology. Future studies should elucidate the molecular interplay between Rap1a and Epac in hypertension and evaluate pharmacological interventions. These findings may have translational relevance to human hypertensive heart disease, where dysregulation of RAGE and Rap1a signaling contributes to ventricular hypertrophy and interstitial fibrosis, highlighting these pathways as potential therapeutic targets.

## 2. Materials and Methods

### 2.1. Animal Model

The subsequent studies utilized 15–16-week-old C57Bl/6 male wildtype (WT) mice (The Jackson Laboratory; Bar Harbor, ME, USA). Generation of RAGE KO mice has been previously described [31,32]. In summary, RAGE KO mice were created using the Cre-Lox recombination strategy to delete exons 2–7, leading to the loss of global RAGE mRNA and non-functional RAGE signaling. These mice were then crossbred with C57Bl/6 mice to produce the RAGE KO cohort used in this study. Additionally, Rap1a knockout (Rap1a KO) mice were also used in this study. The Rap1a KO model was generated by inserting a neomycin resistance gene downstream of exon 4 of RAP1A in the reverse (3′–5′) orientation. To achieve this, a targeting vector containing a 0.95 kb PvuII-NdeI fragment was used to disrupt Rap1a mRNA expression [12,33]. The transgenic mice used in this study are global knockouts. Animals were group-housed in an AAALAC-approved animal facility following the National Institutes of Health “Guide for the Care and Use of Laboratory Animals.” Mice experienced a 12 h/12 h light/dark cycle, and food and water were ad libitum. The University of Mississippi Animal Care and Use Committee (IACUC protocol number 23-011) approved all animal usage protocols. All experiments were performed in male mice. Sex differences in RAGE and Rap1a signaling may influence remodeling outcomes and will be addressed in future studies. All experimental procedures in this study were conducted in accordance with the ARRIVE 2.0 Guidelines to ensure transparency, reproducibility, and ethical rigor in animal research and can be found in the Institutional Review Board Statement below.

### 2.2. Angiotensin II (AngII) Delivery

#### 2.2.1. Preparation of Osmotic Pumps

Osmotic pumps (Alzet model #2004; Alzet, Cupertino, CA, USA) were prepared in a sterile environment following manufacturer’s suggested protocols. Each mouse was weighed to determine the appropriate dose of angiotensin II (AngII; 1 μg/kg/min; Thermo Fisher; Memphis, TN, USA #AAJ6086MB) [12,34,35]. In a laminar flow hood, the required amount of AngII was dissolved in sterile saline in plastic conical tubes. The solution was then aliquoted into sterile microcentrifuge tubes according to mouse body weight and labeled appropriately. A 1 mL sterile syringe fitted with a pump filling needle was used to aspirate the AngII solution, ensuring no air bubbles were introduced. The needle was carefully inserted into the pump body, and the solution was dispensed until a bead of liquid appeared at the pump’s opening. The flow moderator was then inserted to seal the pump, with excess fluid blotted to prevent leakage. A filling ratio of ≥100% was considered acceptable; pumps with ratios < 95% were refilled to eliminate potential air bubbles. The prepared pumps were placed in labeled tubes containing sterile saline and incubated at 37 °C for at least 12 h to allow for partial priming prior to implantation so that AngII infusion commenced approximately 24 h post-implantation.

#### 2.2.2. Pump Implantation Procedure

Mice were anesthetized using 1–4% isoflurane in a vaporizer chamber until loss of the righting reflex, corneal reflex, and tail/paw pinch response was confirmed. The surgical site (midscapular region) was shaved, and the mouse was maintained under 2.5% isoflurane via a nose cone. Artificial tears were applied to prevent corneal desiccation, and the surgical site was sterilized. A sterile drape was placed to expose only the surgical site. A 5 mm midscapular incision was made, and a subcutaneous pocket was created using hemostats. The filled osmotic pump was inserted into the pocket, ensuring a 1 cm distance from the incision site to minimize interference with wound healing. The incision was closed using 5-0 nylon sutures and wound clips. A topical analgesic was applied to the surgical site before the mouse was transferred to a warmed recovery cage with access to water. Mice were monitored post-surgery until they regained righting reflex and were alert. Post-operative care included housing the mice in clean cages with food, water, and 2 mg/mL acetaminophen supplementation as per protocol.

#### 2.2.3. Study Completion

After a 14-day infusion period, mice were euthanized by CO_2_ inhalation followed by cervical dislocation. Blood was collected for plasma analysis, and the heart was excised and sectioned for 4% paraformaldehyde fixation or frozen with dry ice and stored at −80 °C for further analysis.

### 2.3. Echocardiographic Assessments of Left Ventricle

Echocardiographic assessments were conducted prior to pump implantation (day 0) and again two weeks post-implantation (day 14) using the VEVO 3100 Preclinical Imaging System (FUJIFILM VisualSonics, Inc., Toronto, ON, Canada). This system was equipped with two-dimensional (2D) parasternal short- and long-axis brightness mode (B-mode), 2D targeted motion-mode (M-mode), and Doppler-based flow velocity mode to evaluate left ventricular (LV) function. The following LV parameters were measured: heart rate (HR, beats per minute), LV ejection fraction (EF%, percentage of blood ejected per beat), LV fractional shortening (FS%, an index of contractility), LV cardiac output (CO, mL/min), and mitral valve (MV) E-wave/A-wave ratio (MV E/A) as a measure of diastolic function, as previously described [15,36]. Mice were anesthetized individually with approximately 1% isoflurane. To maintain consistency in data collection, HR was continuously monitored and kept within a near-unconscious range of 400–550 BPM throughout the procedure. Representative images of M-mode recordings can be found in Appendix A.

### 2.4. Collagen Visualization and Quantification

Hearts from age-matched Rap1a wild-type and Rap1a knockout mice were fixed in 4% paraformaldehyde, embedded in paraffin, and sectioned at 5 µm thickness from the equatorial region of the LV. Sections were stained with Picric Acid Sirius Red F3BA (PASR) to visualize total collagen content [31]. PASR-stained sections were examined under fluorescent light with triggered exposure, exploiting the birefringent properties of the stain to distinguish collagen fibers (FITC 488 nm) from background. These images were captured at 20× magnification using a video-based image microscope system (Nikon Eclipse; Nikon Instruments Inc., Melville, NY, USA). Quantitative assessment of collagen content was performed using NIH ImageJ (ImageJ 1.54c) by digital color thresholding using defined RGB wavelength ranges to isolate collagen signal from background tissue. Two threshold phases were applied: Phase 1 (collagen capture): Red 0–250, Green 50–120, Blue 0–255; and Phase 2 (background capture): Red 0–255, Green 0–255, Blue 0–255. Percent collagen content was expressed as the ratio of collagen-positive pixels to total field area for each 20× image field. Approximately 15–35 fields per heart were analyzed, and values were averaged to obtain a single mean percentage per sample [31]. Regions containing large vessels, epicardium, or endocardium were excluded from analysis because these structures contain dense connective tissue that could artificially inflate interstitial collagen measurements [31].

### 2.5. Statistical Analysis

Data are presented as either data are presented as means ± standard deviation (SD) or data are presented as means ± standard error of the mean (SEM) as indicated in figure legends. A post hoc power estimate was derived using ventricular weight-to-body weight (VW/BW) as the primary morphometric endpoint. Based on the pooled saline and AngII VW/BW means across genotypes (0.003606 vs. 0.004666, respectively) and the reported 95% confidence interval for their difference, the standardized effect size (Cohen’s d) was approximately 2.3. Under a simple two-group comparison with *n* = 18 animals per treatment and α = 0.05 (two-tailed), this effect size corresponds to high statistical power (≥0.95) to detect the observed AngII-induced hypertrophy, indicating that the study was adequately powered for this primary morphometric outcome. From the power analysis a *n* = 5-6/genotype/treatment was used. *p* values were indicated on graphs and figure legends. Two-way Repeated Measures (RM) ANOVA with Sidak’s multiple comparisons test post hoc analysis for all morphometric measurement and echocardiography data and a one-way ANOVA with Tukey’s post hoc analysis for collagen data were performed with Graph Prism software, version 8.4.3.

## 3. Results

### 3.1. Morphometric Measurements

Multiple comparisons were conducted to assess morphometric differences between saline and AngII treatments within each genotype (WT, RAGE KO, and RapKO). No significant (NS) differences were observed in body weights (BW) between saline and AngII treatments in any of the genotypes (*p* > 0.05 for all), suggesting that neither RAGE nor Rap1a deletion altered BW in response to AngII (Figure 1A). For ventricular weight (VW), two-way ANOVA revealed a highly significant effect of AngII treatment (*p* < 0.0001), increasing VW compared to saline controls. Post-hoc tests confirmed a significant VW increase in WT mice (*p* < 0.05), but not in RAGE KO (*p* = 0.1831) or RapKO (*p* = 0.9999) mice, indicating that AngII induced cardiac hypertrophy in WT mice while RAGE KO and RapKO mice were resistant (Figure 1B). The ventricular weight-to-body weight (VW/BW) ratio was significantly elevated by AngII treatment (*p* < 0.0001), with mean values of 0.003606 (saline) and 0.004666 (AngII) (mean difference = 0.001060; 95% CI: [−0.001366, −0.0007541]). This hypertrophic effect was consistent across genotypes, except in RapKO mice, where no significant change occurred (*p* = 0.8565), suggesting protection against AngII-induced remodeling (Figure 1C). Blood glucose (BG) analysis showed a borderline treatment effect (*p* = 0.0506), with AngII-treated mice trending toward lower BG (mean = 163.9 mg/dL) than saline-treated mice (mean = 202.5 mg/dL; mean difference = 38.58 mg/dL; 95% CI: [−0.1007, 77.27]). However, this did not reach significance. Genotype (*p* = 0.3082) and genotype–treatment interaction (*p* = 0.8089) were also nonsignificant, indicating a modest, uniform AngII effect on BG across genotypes (Figure 1D).

### 3.2. Heart Rate (HR)

Heart rate (HR; beats per minute-BPM) data measured during echocardiography procedures revealed no significant differences across genotypes (*p* = 0.3961), treatment (*p* = 0.7082), or their interactions (genotype × day *p* = 0.7228; genotype × treatment *p* = 0.5861; day × treatment *p* = 0.1916; genotype × day × treatment *p* = 0.7696). HR measurement comparisons showed similar values across genotypes and saline vs. AngII treatment groups at both Day 0 and Day 14 time points with no significant differences observed. Overall, heart rate changes were not influenced by genotype, treatment, or their interactions. Measurements remained similar across all genotypes throughout the study time points (Table 1).

### 3.3. Systolic and Diastolic Left Ventricular Anterior Wall (LVAW) Thicknesses

Mean systolic LVAW thickness was evaluated to assess genotype and treatment effects. A significant genotype–treatment interaction (*p* = 0.0233) occurred indicating that AngII impacted systolic LVAW, and it varied by genotype. Genotype alone was not significant (*p* = 0.6860), but AngII treatment had a strong effect (*p* = 0.0019), increasing systolic LVAW across genotypes. Thus, AngII is a primary driver of systolic LVAW, with genotype modulating its effect (Figure 2A). For diastolic LVAW thickness, AngII treatment showed a highly significant effect (*p* = 0.0002), markedly altering diastolic values. Genotype alone was nonsignificant (*p* = 0.5099), but the treatment-genotype interaction was significant (*p* = 0.038), indicating genotype-specific responses to AngII. These results underscore AngII-mediated changes in diastolic LVAW, with effects varying by genotype (Figure 2B). Overall, AngII significantly influences both systolic and diastolic LVAW, modulated by genotype-specific interactions.

### 3.4. Systolic and Diastolic Left Ventricular Posterior Wall (LVPW) Thicknesses

Systolic LVPW thickness showed no significant effects from genotype (*p* = 0.7865), AngII treatment (*p* = 0.6976), or their interaction (*p* = 0.3644), suggesting that experimental conditions did not alter this parameter (Figure 3A). Likewise, diastolic LVPW thickness was unaffected by genotype (*p* = 0.7829), AngII treatment (*p* = 0.2232), or their interaction (*p* = 0.3143), indicating no impact from the tested factors (Figure 3B). Although genotype and AngII treatment had no significant effects on either systolic or diastolic LVPW thickness, substantial variance was linked to individual animal differences, highlighting physiological variability as a key influence on these measurements.

### 3.5. End-Systolic and End-Diastolic Left Ventricular (LV) Volumes

Systolic LV volume analysis showed no significant effects of genotype (*p* = 0.1087), AngII treatment (*p* = 0.6334), or their interaction (*p* = 0.4964), suggesting that individual variability, rather than experimental factors, drove the observed data (Figure 4A). Similarly, diastolic LV volume, measured by echocardiography, was unaffected by genotype (*p* = 0.1616), AngII treatment (*p* = 0.1640), or their interaction (*p* = 0.5544) (Figure 4B). When normalized to body weight, end-systolic volume-to-body-weight ratio showed no significant effects from genotype (*p* = 0.1245), AngII treatment (*p* = 0.3429), or their interaction (*p* = 0.4384) (Figure 4C). The end-diastolic volume-to-body-weight ratio was similarly unaffected by genotype (*p* = 0.4460), AngII treatment (*p* = 0.8422), or their interaction (*p* = 0.4479) (Figure 4D). Two-way repeated measures ANOVA across all LV volume measures confirmed no significant effects of genotype, AngII treatment, or their subsequent interactions.

### 3.6. Left Ventricular (LV) Functional Analysis by Echocardiography

Stroke volume (SV) analysis revealed no significant effects of genotype (*p* = 0.3413) or the genotype-AngII interaction (*p* = 0.3025), but AngII treatment significantly increased SV (*p* = 0.0429), indicating its role in this parameter (Figure 5A). Ejection fraction (%EF) showed no significant effects from genotype (*p* = 0.1579), AngII treatment (*p* = 0.1120), or their interaction (*p* = 0.3666) (Figure 5B). Fractional shortening (%FS) was similarly unaffected (genotype: *p* = 0.1157; AngII: *p* = 0.1077; interaction: *p* = 0.5672) (Figure 5C). Cardiac output (CO), assessed by two-way repeated measures ANOVA, showed no significant effects (genotype: *p* = 0.2020; AngII: *p* = 0.3859; interaction: *p* = 0.3735) (Figure 5D). Across SV, %EF, %FS, and CO, only AngII’s effect on SV was significant (*p* = 0.0429), while other parameters remained unchanged (*p* > 0.05). This suggests that, apart from SV, genotype and AngII treatment had minimal impact on cardiac function under these conditions, with AngII influencing cardiac efficiency via changes in SV.

### 3.7. Mitral Valve (MV) Flow Parameters

Early-phase ventricular filling velocity (E wave; MV E; mm/s) showed no significant effects of genotype (*p* = 0.1551), AngII treatment (*p* = 0.0804), or their interaction (*p* = 0.4380), indicating that these factors did not substantially alter LV diastolic function (Figure 6A). In contrast, late-phase filling (A wave; MV A; mm/s) was significantly increased by AngII treatment (*p* = 0.0182), suggesting an effect by AngII on LV function. Genotype trended toward significance (*p* = 0.0610) but remained nonsignificant, and the genotype-treatment interaction was not significant (*p* = 0.4938). This implies AngII modulates MV A, possibly via blood pressure or remodeling, with minimal genotype influence (Figure 6B). The MV E/A ratio, a marker of diastolic function, showed no significant effects from genotype (*p* = 0.2662), AngII treatment (*p* = 0.5643), or their interaction (*p* = 0.2095), suggesting no substantial impact on this parameter (Figure 6C).

### 3.8. Myocardial Collagen Percentage

Quantitative analysis of PASR-stained cardiac sections revealed significant differences in collagen content across genotype and treatment groups (Figure 7). AngII infusion in WT mice significantly increased collagen deposition compared to saline-treated RAGE KO and RapKO mice (WT AngII vs. RAGE KO Saline: *p* = 0.0049; WT AngII vs. RapKO Saline: *p* = 0.0010) and compared to AngII-treated RapKO mice (*p* = 0.0092). The collagen content in WT AngII treated hearts was nearly double that of WT Saline treated hearts, though this difference did not reach statistical significance (*p* = 0.0621). RAGE KO and RapKO mice exhibited a markedly blunted fibrotic response to AngII infusion, with mean collagen levels of 2.54% and 1.65%, respectively, compared to 3.09% in WT AngII mice. Notably, RAGE KO AngII hearts displayed higher collagen content than RapKO AngII, suggesting RapKO mice demonstrated near-complete suppression of AngII-induced collagen accumulation. No significant differences were detected among saline-treated groups, indicating that genotype alone did not affect baseline collagen deposition. Overall, these data demonstrate that both RAGE and Rap1a deficiency mitigate AngII-induced cardiac fibrosis. The strongest differences were observed between WT AngII and RapKO Saline or AngII groups (*p* < 0.01–0.001)**,** supporting the hypothesis that RAGE–Rap1a signaling contributes substantially to AngII-mediated collagen remodeling in the myocardium.

### 3.9. Summary of Results

The study examined the effects of AngII treatment on various cardiac and metabolic parameters across different genotypes (WT, RAGE KO, and RapKO). Body weight remained unchanged regardless of genotype or treatment. AngII significantly increased ventricular weight and the VW/BW ratio, indicating cardiac hypertrophy, but RAGE KO and RapKO mice were protected. AngII treatment showed a trend toward lowering blood glucose levels, though not significantly. Heart rate was unaffected by genotype, treatment, or time. Ventricular wall thickness measurements revealed AngII treatment-dependent and genotype-specific effects, particularly for LVAW diastolic thickness, which exhibited the most pronounced changes. AngII-induced cardiac remodeling was primarily driving changes in ventricular structure. Graphically genotype influenced systolic and diastolic volumes, though there was no significant effect. Additionally, AngII treatment significantly increased myocardial collagen deposition in WT mice, whereas both RAGE and Rap1a deficiency markedly attenuated this fibrotic response. Overall, the results highlight the role of genotype in modulating the hypertrophic and fibrotic response to AngII, with RapKO mice showing resistance to these effects.

## 4. Discussion

This study advances our understanding of the molecular pathways underlying hypertension and cardiac remodeling, focusing on the roles of RAGE and Rap1a in AngII-mediated effects [1,2,3]. Hypertension drives cardiovascular disease through maladaptive remodeling, with RAGE, Rap1a, and AngII interacting within a complex signaling network that promotes hypertrophy, inflammation, and fibrosis [4,5,6].

AngII, a key effector of RAAS, exerts its hypertrophic effects AT_1_R, activating MAPKs and PI3K/Akt pathways to enhance cardiomyocyte growth, oxidative stress, and matrix remodeling [7,8]. RAGE and AT_1_R have both been implicated in pathological processes associated with inflammation, oxidative stress, and cardiovascular dysfunction. Evidence has begun to suggest these receptors interact functionally [8,9]. Additionally, AT_1_R activation has been shown to enhance RAGE expression, while RAGE activation can conversely upregulate components of the RAAS, including AT_1_R, establishing a feed-forward loop that exacerbates cardiovascular complications [8]. Prior studies demonstrate that inhibiting either receptor reduces remodeling, suggesting that dual targeting of the RAGE-AT_1_R axis could be a viable therapeutic approach [10,11,12].

Rap1a modulates endothelial and vascular smooth muscle function, fibroblast activity, and inflammation, regulating vascular tone and permeability via integrin and barrier mechanisms [13,14,15,17]. Dysregulation of Rap1a has been associated with endothelial dysfunction, a hallmark of hypertension [19,20,21]. In a study by Xie et al., AngII modulates inflammatory cytokine expression (e.g., IL-1β, IL-6, IL-8, TNF-α) in tubular cells through a pathway involving Rap1a as a key intermediary [22,23]. While AngII treatment has been shown to cause a dose-dependent decrease in Rap1 activity, the expression and translocation of NHE3, a sodium/hydrogen exchanger, was altered through cAMP-dependent pathways [22]. Similarly, Epac1 and Epac2, important downstream effectors of cAMP, have gained attention for their role in cardiovascular physiology and pathology. Epac isoforms mediate a variety of cellular processes, including cardiac hypertrophy and fibrosis [16,28,37]. Studies have shown that activation of Epac1 can lead to maladaptive hypertrophic remodeling by promoting extracellular matrix deposition in the lungs as well as in cardiac myofibroblast collagen gel contraction [13,24,25,38]. While Epac2 has been reported to have protective effects in certain cardiac contexts, such as reducing apoptosis under stress conditions [26]. The dual nature of the Epac isoforms underscores their complexity in the hypertrophic response, necessitating further investigation.

Our results support the hypothesis that RAGE and Rap1a modulate AngII-induced hypertensive responses. AngII increased VW and VW/BW ratio in WT and RAGE KO mice, indicating hypertrophy, whereas RapKO mice were protected, suggesting Rap1a’s role in maladaptive remodeling. Body weight remained unchanged across groups, and AngII showed a nonsignificant trend toward lowering blood glucose, consistent with reports of enhanced insulin sensitivity [39,40,41,42]. Heart rate was unaffected by genotype or treatment.

Echocardiographic data revealed that AngII-driven structural remodeling primarily affected LVAW and LVPW thickness with treatment-dependent changes most evident in diastolic LVAW. Genotype influenced specific outcomes, notably in RAGE KO and RapKO mice, while LV volumes showed genotype-dependent variation but were largely unaltered by AngII. RapKO mice’s resistance to hypertrophy reinforces Rap1a’s contribution to remodeling. Left ventricular function was shaped by AngII, with SV and CO significantly altered by treatment, reflecting hemodynamic effects rather than contractility changes (e.g., %EF, %FS) [43,44]. Genotype influenced %FS, aligning with genetic effects on myocardial mechanics [44], but these effects were stable in response to treatment. While AngII elevates systemic pressure, telemetry data needs to be performed to confirm comparable hemodynamic loads among genotypes in order to determine genotype-specific responses arising from molecular signaling differences rather than mechanical overload.

Diastolic function, assessed MV E and MV A, varied by genotype, indicating genetic impacts on filling dynamics. However, the absence of genotype–treatment interaction suggests these differences persisted independently of AngII. The MV E/A ratio remained unchanged, suggesting compensatory mechanisms balanced early and late filling phases. These findings parallel patterns seen in human hypertensive remodeling, where increased RAGE and Rap1a signaling correlate with fibrosis severity. Targeting these pathways could thus have therapeutic implications.

This study also demonstrated that AngII induced significant myocardial collagen accumulation, and both RAGE and Rap1a deficiency confer protection against this fibrotic response. Consistent with prior studies from our laboratory linking AGE/RAGE signaling to maladaptive cardiac remodeling, our data showed that genetic deletion of either RAGE or Rap1a attenuates collagen deposition, indicating that the RAGE–Rap1a axis plays a pivotal role in AngII-driven fibrosis. The observed increase in collagen content in WT AngII hearts aligns with classical models of pressure overload, where fibroblast activation and extracellular matrix remodeling contribute to ventricular stiffening [45]. In contrast, RKO and RapKO hearts exhibited markedly reduced collagen accumulation despite AngII infusion, suggesting that interruption of RAGE signaling or its downstream mediator Rap1a blunts profibrotic signaling cascades. These findings indicate that RAGE–Rap1a signaling promotes collagen synthesis but may also be impacted by AGE-mediated crosslinking, compounding myocardial stiffness and impairing compliance. Together, these results expand on prior work demonstrating that RAGE activation enhances extracellular matrix remodeling and fibroblast activation under hypertensive conditions. The protective phenotype observed in RAGE KO and RapKO mice suggested that RAGE and Rap1a may function cooperatively to amplify downstream signaling via ERK and PKC-ζ pathways, both of which are known to regulate myofibroblast differentiation and collagen secretion. Future studies will evaluate downstream activation of Epac1, PKC-ζ, and MAPK signaling as mediators of Rap1a- and RAGE-dependent remodeling. Ongoing work will assess physical interactions between Rap1a and RAGE through co-immunoprecipitation and phosphorylation analyses to define their direct regulatory link. Pharmacologic agents such as RAGE antagonists (FPS-ZM1, azeliragon), Epac1 inhibitors (ESI-09), and Rap1a activators (8-pCPT-2′-O-Me-cAMP) may mimic or enhance the protective effects observed in knockout models. Future studies will extend these findings to include female mice, as sexual dimorphism in RAGE and Rap1a signaling may influence hypertensive remodeling outcomes. Estrogen and other sex hormones are known to modulate RAAS activity, oxidative stress, and extracellular matrix turnover, which could alter the molecular responses observed in this study. Inclusion of female cohorts will therefore be critical to fully delineate sex-specific mechanisms and improve translational relevance to human cardiovascular disease. The sample size estimates support the adequacy of the sample size for the main AngII effect on VW/BW, although larger cohort sizes and extending the AngII infusion length of time will be important for subtler genotype-specific or interaction effects.

## 5. Conclusions

Our data align with prior research demonstrating that AngII primarily influences hemodynamic parameters such as SV and CO, rather than directly altering myocardial contractility [1,6,44]. Our findings suggest that alterations in Rap1a signaling may mitigate cardiovascular remodeling in an AngII-induced hypertensive model. This aligns with previous research highlighting the cAMP-EPAC-Rap1a axis as a critical regulator of cardiovascular function and myocardial adaptation [1,22,24,28]. Targeting Rap1a activation or inhibiting Epac1 could offer therapeutic potential for hypertensive cardiac remodeling. Future studies should investigate whether pharmacological modulation of these pathways improves cardiovascular outcomes and further elucidate the molecular interactions between Rap1a and Epac1 in hypertensive models. Collectively, these results identify Rap1a and RAGE as interrelated regulators of hypertensive cardiac remodeling and potential therapeutic targets for human heart failure with preserved ejection fraction (HFpEF). Future pharmacological studies will be critical for clinical translation.

## Figures and Tables

**Figure 1 cells-14-01834-f001:**
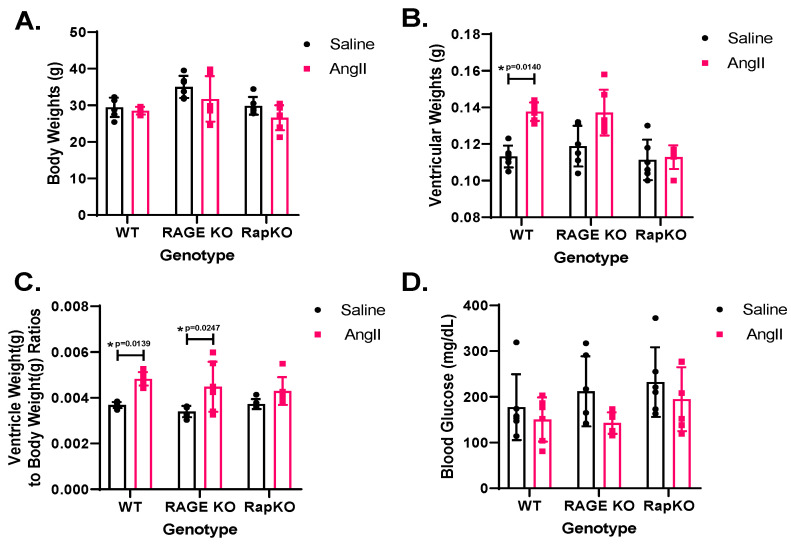
(**A**) No significant differences were observed between Saline and AngII treatments in any genotype. (**B**) Significant differences in ventricular weights (VW) were observed between Saline and AngII treated mice for the WT genotype. There were no significant differences observed in RAGE KO and RapKO mice. (**C**) When ventricular weights (VW) were normalized to body weight (BW) significant differences were observed between Saline and AngII treated mice in WT and RAGE KO genotypes. There were no significant differences observed in RapKO mice. (**D**) AngII reduced blood glucose in all genotypes; however, these differences were not significantly different from Saline treated mice. Data are presented as means ± standard deviation (SD). *n* = 6/genotype/treatment. *p* values were indicated on graphs; * *p* < 0.05; Saline vs. AngII treatment within genotype. *p* values were determined by 2-way ANOVA with Sidak’s multiple comparisons test post hoc analysis.

**Figure 2 cells-14-01834-f002:**
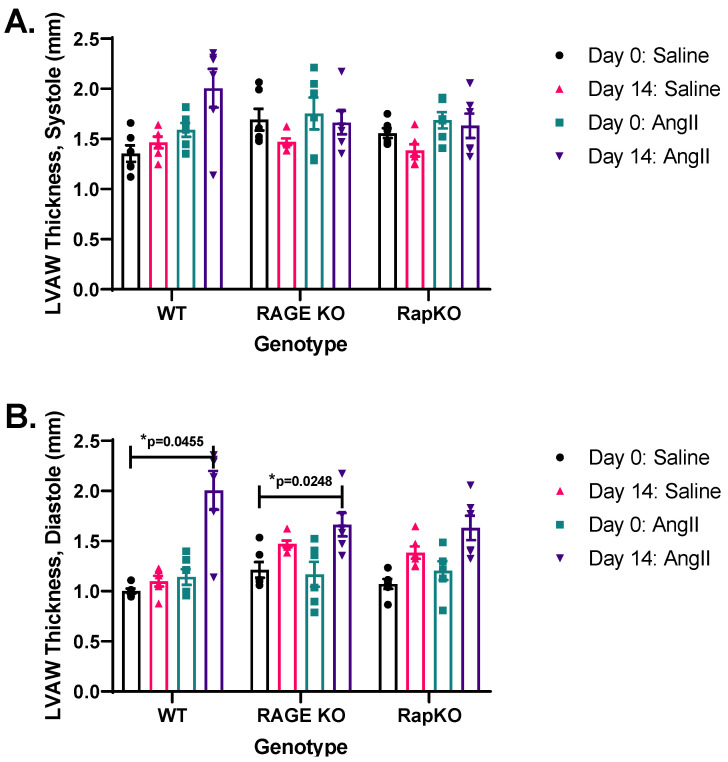
Two-way RM ANOVA analyses revealed that AngII significantly influenced both (**A**) systolic (*p* = 0.0019) and (**B**) diastolic (*p* = 0.0002) left ventricular anterior wall (LVAW) measurements. For systolic values, there was a significant genotype × treatment interaction (*p* = 0.0233). Similarly, diastolic measurements showed a significant interaction (*p* = 0.0381), suggesting differential responses across conditions. These findings highlight AngII treatment as the primary determinant of LVAW function, with variations depending on genotype. Data are presented as means ± standard error of the mean (SEM). *n* = 6/genotype/treatment. *p* values were indicated on graphs. * *p* < 0.05. Two-way RM ANOVA with Sidak’s multiple comparisons test post hoc analysis was performed. Representative images of M-mode recordings can be found in Appendix A.

**Figure 3 cells-14-01834-f003:**
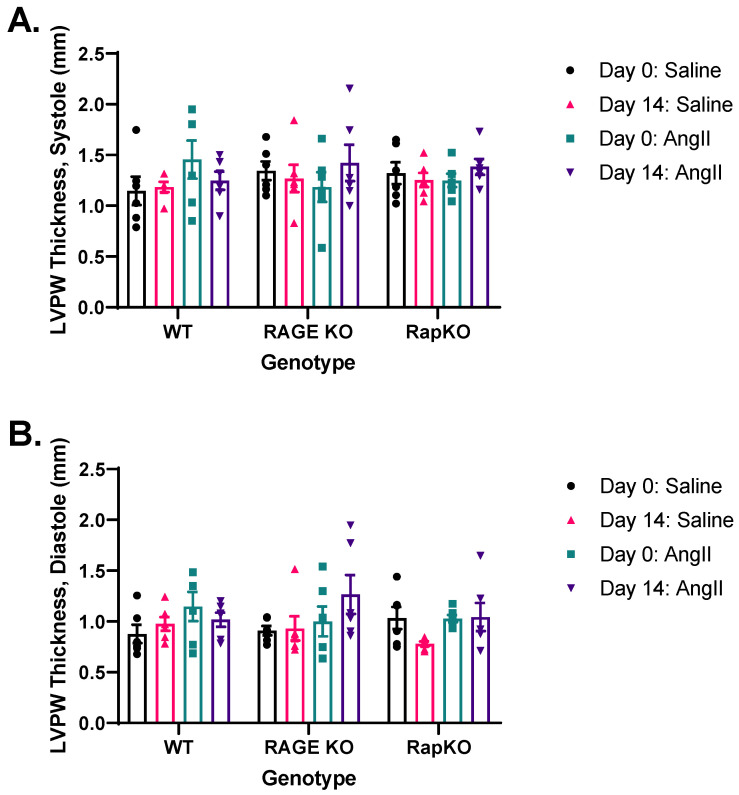
Analysis of left ventricular posterior wall (LVPW) measurements was conducted. (**A**) Systolic LVPW measurements showed no significant effects of the genotype (*p* = 0.7865), AngII treatment (*p* = 0.6976), or their interaction (*p* = 0.3644). (**B**) Diastolic LVPW measurements similarly exhibited no significant main effects for genotype (*p* = 0.7829), AngII treatment (*p* = 0.2232), or their interaction (*p* = 0.3143). Data are presented as means ± standard error of the mean (SEM). *n* = 6/genotype/treatment. Two-way RM ANOVA with Sidak’s multiple comparisons test post hoc analysis was performed. No significance was found between genotypes or treatment. Representative images of M-mode recordings can be found in Appendix A.

**Figure 4 cells-14-01834-f004:**
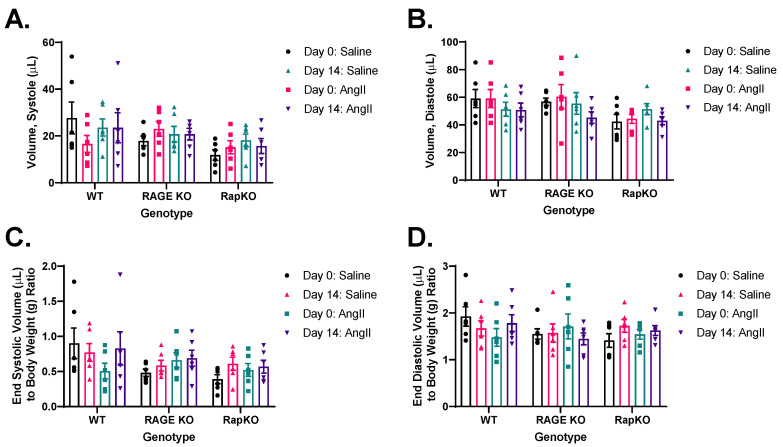
Analysis of the effects of genotype, AngII treatment, and their interaction on (**A**) systolic volume, (**B**) diastolic volume, (**C**) end-systolic volumes adjusted for body weight and (**D**) end-diastolic volumes adjusted for body weight. No significant effects were found for genotype (*p* > 0.05), AngII treatment (*p* > 0.05), or the genotype × AngII treatment interaction (*p* > 0.05) in any analysis. Data are presented as means ± standard error of the mean (SEM). *n* = 6/genotype/treatment. Two-way RM ANOVA with Sidak’s multiple comparisons test post hoc analysis was performed. No significance was found between genotypes or treatments. Representative images of M-mode recordings can be found in Appendix A.

**Figure 5 cells-14-01834-f005:**
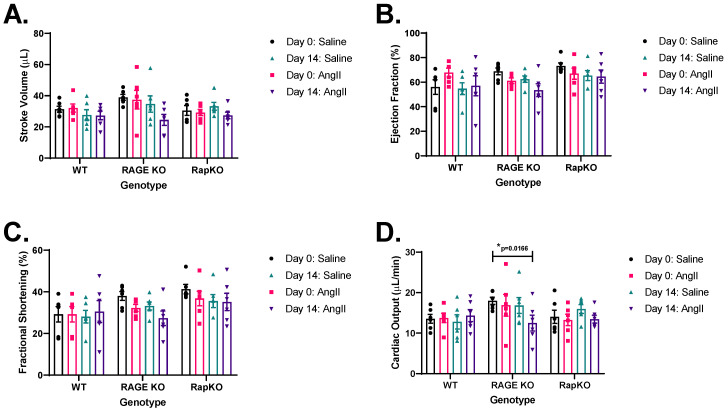
The results from echocardiographic functional analysis of the left ventricle for (**A**) stroke volume (SV), (**B**) ejection fraction (%EF), (**C**) fractional shortening (%FS), and (**D**) cardiac output (CO). No significant effects were found for genotype, AngII treatment, or their interaction in the ejection fraction, fractional shortening, and cardiac output analyses (*p* > 0.05). Genotype in the SV analysis was the only factor showing a significant effect (*p* = 0.0429). Data are presented as means ± standard error of the mean (SEM). *n* = 6/genotype/treatment. *p* values were indicated on graphs; *p* > 0.05; Saline vs. AngII treatment within genotype. *p* values were determined by 2-way RM ANOVA with Sidak’s multiple comparisons test post hoc analysis. Representative images of M-mode recordings can be found in Appendix A.

**Figure 6 cells-14-01834-f006:**
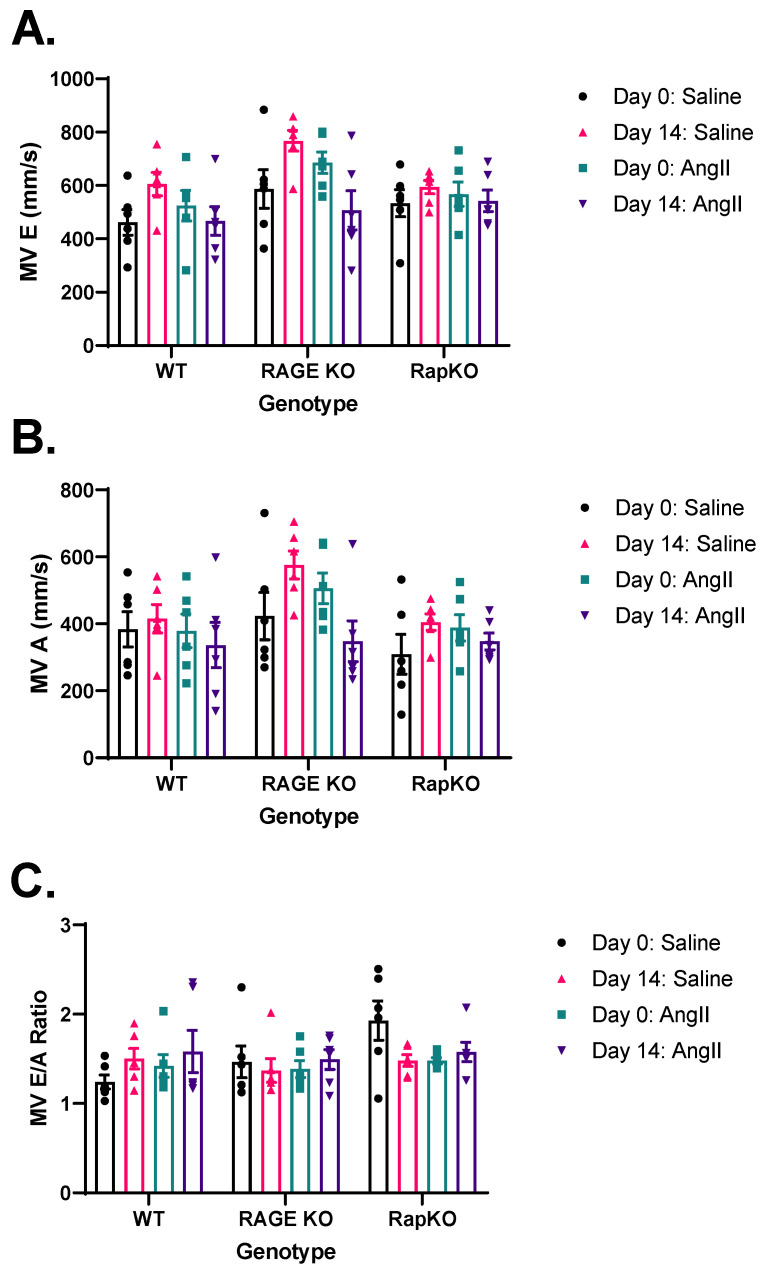
Analysis of left ventricular (LV) function showed (**A**) MV E (mm/s) had no significant effects of genotype, AngII treatment, or their interaction. (**B**) MV A (mm/s) was significantly influenced by AngII treatment (*p* = 0.0182), while genotype and their interaction had no significant effects. (**C**) MV E/A ratio also showed no significant effects from either factor or their interaction on LV function. These findings suggest that AngII treatment plays a significant role in modulating LV function, independent of genotype. Data are presented as means ± standard error of the mean (SEM). *n* = 6/genotype/treatment. *p* values were indicated on graphs; Saline vs. AngII treatment within genotype. *p* values were determined by 2-way ANOVA with Sidak’s multiple comparisons test post hoc analysis. No significance was found between genotypes or treatment. Representative images of M-mode recordings can be found in Appendix A.

**Figure 7 cells-14-01834-f007:**
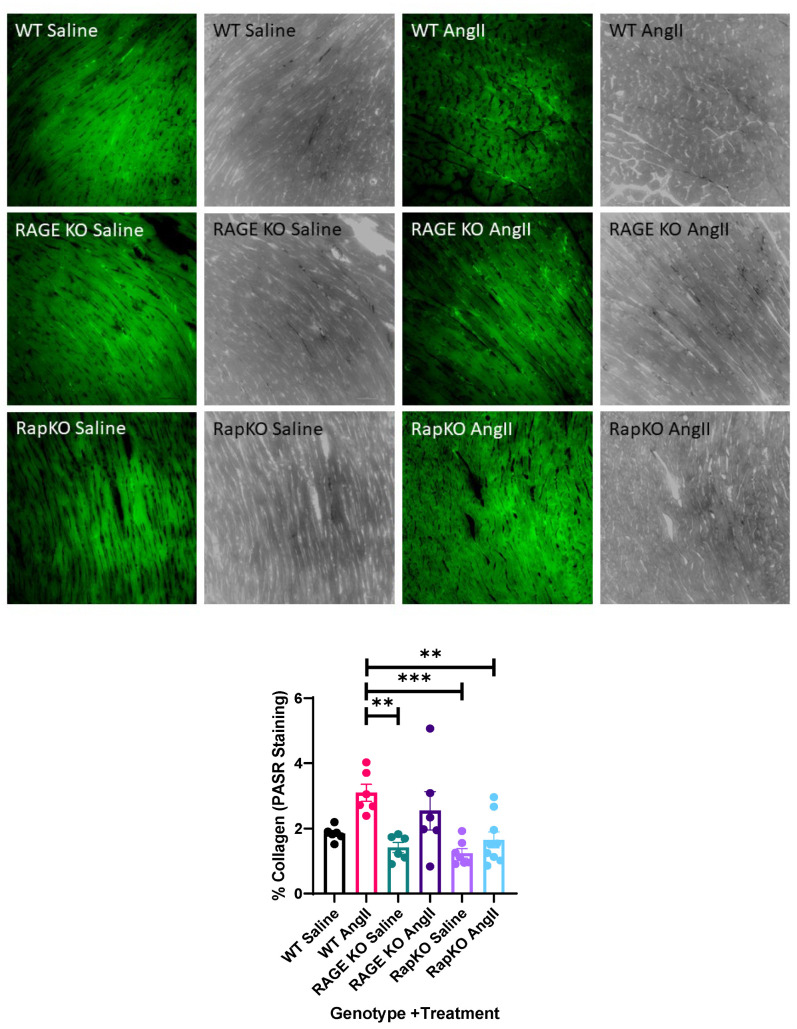
Representative 20× fluorescence images showing total collagen (FITC, 488/520 nm; green) in left-ventricular sections from WT, RAGE KO, and RapKO mice treated with saline or AngII. Sections were fixed in 4% paraformaldehyde and imaged under identical exposure settings using a Nikon Eclipse fluorescence microscope (Nikon Instruments Inc., Melville, NY, USA) using triggered FITC exposure and acquisition with a digital CMOS camera (Nikon Instruments Inc., Melville, NY, USA). Bright field images of PASR were also captured at 20×. Collagen content was quantified in ImageJ as percent fluorescent area per field (15–35 fields per heart; scale bar = 100 µm or 20 µm). Data are presented as means ± standard error of the mean (SEM); *p* values were determined by 1 way ANOVA with Tukey’s post hoc analysis ** *p* < 0.01, *** *p* < 0.001.

**Table 1 cells-14-01834-t001:** This table summarizes heart rate (HR; beats per minute—BPM) data across the different genotypes (WT, RAGE KO, and RapKO), treatments (saline and AngII), and time points (Day 0 and Day 14). HR did not differ by genotype (*p* = 0.8840) or treatment (*p* = 0.3920), with no significant interaction (*p* = 0.4083). Data are presented as means ± standard error of the mean (SEM). *n* = 6/genotype/treatment. Two-way ANOVA with Sidak’s multiple comparisons test post hoc analysis was performed. No significance was found between genotypes or treatment.

Genotype	Day 0: Saline Mean (SEM)	Day 0: AngII Mean (SEM)	Day 14: Saline Mean (SEM)	Day 14: AngII Mean (SEM)
WT	430.43 (16.91)	429.30 (19.82)	459.79 (28.48)	530.69 (30.36)
RAGE KO	463.76 (20.41)	454.99 (18.70)	498.08 (23.26)	517.49 (46.64)
RapKO	460.68 (23.61)	447.39 (22.06)	483.61 (25.52)	494.93 (17.99)

## Data Availability

Data supporting reported results can be found at. https://doi.org/10.6084/m9.figshare.28733219 (accessed on 1 November 2025).

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
