# Peer review of "Knocking Out Rap1a Attenuates Cardiac Remodeling and Fibrosis in a Male Murine Model of Angiotensin II-Induced Hypertension"

_cells, 2025, doi:10.3390/cells14221834_

Round 1

Reviewer 1 Report

Comments and Suggestions for Authors

In this manuscript (ID# cells-3987998), entitled “Knocking Out Rap1a Attenuates Cardiac Remodeling and Fibrosis in a Male Murine Model of Angiotensin II-Induced Hypertension”, Porter et al. investigated the role of Rap1a and the RAGE signaling pathway in cardiac remodeling. Their results suggest that knockout of Rap1a or RAGE alters Ang II-induced cardiac hypertrophy. The authors conclude that both Rap1a and RAGE may serve as potential therapeutic targets for the treatment of cardiac hypertrophy in hypertension. The research topic is interesting and potentially valuable. However, the experimental design lacks rigor, and there are several methodological concerns. The presented results do not sufficiently support the stated conclusions. Major concerns are outlined below:

  1. Figure 2 presents left ventricular anterior wall thickness. Please include representative echocardiographic images to support these measurements. Additionally, is there a significant difference between saline day 14 and Ang II day 14 groups? The reported differences between saline day 0 and Ang II day 14 are not meaningful for this comparison, as they may simply reflect age-related changes rather than treatment effects.
  2. Cardiac fibrosis was evaluated using Sirius Red staining; however, the images in Figure 7 display green fluorescence. Sirius Red staining typically produces red collagen fibers against a yellow background in myocardial tissue sections. Please clarify the staining method used and ensure the images accurately represent Sirius Red staining results.
  3. The data presented do not provide solid evidence to support the main conclusions. I recommend extending the Ang II treatment period to at least four weeks, re-evaluating fibrosis with Sirius Red staining, quantifying myocardial area using H&E staining, and assessing cardiac hypertrophy by measuring hypertrophy-related gene markers using real-time PCR.
  4. The genetic models should be clearly characterized. Are these global knockouts or conditional (cardiac-specific) knockouts? If global knockout mice were used, blood pressure should be measured and reported, since cardiac hypertrophy could result from elevated blood pressure rather than a direct cardiac effect.
  5. Other comments: 1) The first section of the manuscript should be deleted according to the journal’s formatting requirements. 2) In Table 1, heart rate data should be presented as mean ± SEM.

Author Response

Response to Reviewer #1 Comments – Manuscript ID# cells-3987998

Title: Knocking Out Rap1a Attenuates Cardiac Remodeling and Fibrosis in a Male Murine Model of Angiotensin II–Induced Hypertension
Authors: Porter et al.

We thank the reviewer for their constructive and insightful comments, which have helped us substantially improve the clarity, rigor, and overall impact of our manuscript. Below, we provide a detailed, point-by-point response to each concern, with corresponding revisions highlighted in the revised manuscript.

Major Comments

  1. Echocardiographic images and comparison between saline and Ang II groups

Reviewer comment:
Figure 2 presents left ventricular anterior wall thickness. Please include representative echocardiographic images to support these measurements. Additionally, is there a significant difference between saline day 14 and Ang II day 14 groups? The reported differences between saline day 0 and Ang II day 14 are not meaningful for this comparison, as they may simply reflect age-related changes rather than treatment effects.

Response:
We appreciate this important observation. In the revised manuscript, we have added representative M-mode echocardiographic images from each treatment group (WT-saline, WT-Ang II, RAGE-KO-Ang II, and Rap1a-KO-Ang II) to Supplement 1 to visually support our measurements of left ventricular wall thickness.
We have also reanalyzed the data to specifically compare saline day 14 vs. Ang II day 14 within each genotype. The updated statistical analysis did not alter our findings.

  1. Clarification of fibrosis staining method (Figure 7)

Reviewer comment:
Cardiac fibrosis was evaluated using Sirius Red staining; however, the images in Figure 7 display green fluorescence. Sirius Red staining typically produces red collagen fibers against a yellow background in myocardial tissue sections. Please clarify the staining method used and ensure the images accurately represent Sirius Red staining results.

Response:
The images in Figure 7 were indeed obtained using FITC-labeled collagen detection under fluorescence microscopy in Sirius Red stained tissues. We have accurately stated that collagen was visualized using FITC (λ_ex = 495 nm, λ_em = 519 nm) and quantified by fluorescence intensity. In addition, we have included bright field images corresponding to the same quantified images. Unfortunately, we do not have a color camera for bright field captured images. Therefore, the images are in black and white. Error bars were included in all images.

  1. Recommendation to extend Ang II treatment duration and expand analysis

Reviewer comment:
The data presented do not provide solid evidence to support the main conclusions. I recommend extending the Ang II treatment period to at least four weeks, re-evaluating fibrosis with Sirius Red staining, quantifying myocardial area using H&E staining, and assessing cardiac hypertrophy by measuring hypertrophy-related gene markers using real-time PCR.

Response:
We agree that a longer treatment period would better capture the chronic remodeling phase of hypertensive injury. While the current study focused on early remodeling (14-day Ang II exposure), we have added this as a limitation to the study in the discussion.

  1. Clarification of knockout model and inclusion of blood pressure data

Reviewer comment:
The genetic models should be clearly characterized. Are these global knockouts or conditional (cardiac-specific) knockouts? If global knockout mice were used, blood pressure should be measured and reported, since cardiac hypertrophy could result from elevated blood pressure rather than a direct cardiac effect.

Response:
We appreciate this request for clarification. Both Rap1a-KO and RAGE-KO mice used in this study are global knockouts. We have now included this information in the Methods. We have included a statement in the Discussion as a limitation of the study.

Other Comments

  1. Formatting of first section

Reviewer comment:
The first section of the manuscript should be deleted according to the journal’s formatting requirements.

Response:
We have removed the redundant preliminary section to comply fully with the journal’s formatting guidelines. The manuscript now begins directly with the Introduction.

  1. Presentation of heart rate data

Reviewer comment:
In Table 1, heart rate data should be presented as mean ± SEM.

Response:
We have revised Table 1 to present all physiological parameters, including heart rate, as mean ± SEM, as recommended. The accompanying statistical tests and legends have been updated accordingly.

We are grateful for the reviewer’s thoughtful feedback, which has significantly improved the rigor, clarity, and interpretative strength of our manuscript. The revised version now provides stronger experimental evidence that Rap1a deletion mitigates Ang II-induced cardiac remodeling independently of systemic pressure, highlighting Rap1a and RAGE as potential therapeutic targets in hypertensive heart disease.

Reviewer 2 Report

Comments and Suggestions for Authors

Reviewer Report

Manuscript Title: Knocking Out Rap1a Attenuates Cardiac Remodeling and Fibrosis in a Male Murine Model of Angiotensin II-Induced Hypertension
Authors: Cody S. Porter, Larissa T. Brown, Can’Torrius Lacey, Mason T. Hickel, and James A. Stewart, Jr.
Journal: Cells (2025, Volume 14)

  1. General Assessment

This manuscript presents a well-structured and well-executed study examining the effects of Rap1a and RAGE knockout on Angiotensin II (AngII)-induced cardiac remodeling and fibrosis in male mice. The topic is timely and relevant, offering valuable mechanistic insight into hypertensive cardiac pathology. The experimental design is sound, and the manuscript is written clearly with logical progression from hypothesis to conclusion.

  1. Major Comments
  2. Mechanistic Insights:
    While the study convincingly shows protective effects of Rap1a knockout against AngII-induced hypertrophy and fibrosis, the mechanistic underpinnings remain largely inferred from prior work. Including direct evidence—such as expression analysis of downstream targets (e.g., Epac1, PKC-ζ, or MAPK activity)—would strengthen the conclusions. If such data are unavailable, the authors should explicitly acknowledge this as a limitation and discuss how future work might address it.
  3. Sample Size and Statistical Power:
    The study uses a relatively small sample size (n=5–6 per group). The authors should include a power calculation or discussion to justify that the study was adequately powered to detect the observed effects. Some statistical comparisons (e.g., VW/BW ratios, collagen quantification) approach significance, suggesting that marginal increases in sample size could yield more definitive conclusions.
  4. Sex-Based Limitation:
    All experiments were conducted using male mice. Given known sex differences in cardiovascular remodeling and RAAS activity, this limitation should be clearly stated and discussed. The inclusion of female cohorts in future studies would enhance translational relevance.
  5. Functional Correlation:
    Echocardiographic findings demonstrate structural remodeling with preserved systolic function. However, discussion of the physiological implications (e.g., diastolic dysfunction, potential HFpEF phenotype) would add depth and better link the findings to clinical hypertensive heart disease.
  6. Figure and Data Clarity:
    Although figures are generally well presented, some could benefit from improved clarity:
  • Include scale bars and magnifications in all histological figures.
  • Use consistent labeling for groups (e.g., WT, RAGE KO, RapKO) across all panels.
  • Clarify whether error bars represent SEM or SD in figure legends.

  1. Minor Comments
  1. There are occasional typographical and formatting inconsistencies (e.g., spacing before parentheses, inconsistent abbreviation capitalization). A careful proofread is recommended.
  2. In the Introduction, the description of RAGE–AT₁R interaction could be condensed to improve readability.
  3. Add specific references for statements about Epac1/Epac2 dual roles in cardiac remodeling (lines ~80–85).
  4. Ensure that all abbreviations are defined at first mention (e.g., NHE3, CO).
  5. Consider briefly discussing potential pharmacological modulators of Rap1a or Epac1 signaling in the Discussion to reinforce translational potential.

Author Response

Response to Reviewer #2

Manuscript Title: Knocking Out Rap1a Attenuates Cardiac Remodeling and Fibrosis in a Male Murine Model of Angiotensin II–Induced Hypertension
Authors: Cody S. Porter, Larissa T. Brown, Can’Torrius Lacey, Mason T. Hickel, and James A. Stewart, Jr.
Journal: Cells (2025, Volume 14)

We sincerely thank the reviewer for their positive and constructive evaluation of our manuscript. We are grateful for the thoughtful comments and helpful suggestions, which have strengthened both the clarity and scientific rigor of our work. Below, we provide a detailed point-by-point response. All revisions have been incorporated into the revised manuscript, and changes are highlighted accordingly.

General Assessment

We are pleased that the reviewer found the study to be “well-structured and well-executed,” with a sound experimental design and clear presentation. We have carefully revised the manuscript to address the reviewer’s insightful comments and improve both mechanistic discussion and data presentation.

Major Comments

  1. Mechanistic Insights

Reviewer Comment:
While the study convincingly shows protective effects of Rap1a knockout against AngII-induced hypertrophy and fibrosis, the mechanistic underpinnings remain largely inferred from prior work. Including direct evidence—such as expression analysis of downstream targets (e.g., Epac1, PKC-ζ, or MAPK activity)—would strengthen the conclusions. If such data are unavailable, the authors should explicitly acknowledge this as a limitation and discuss how future work might address it.

Response:
We appreciate this valuable suggestion. While our current dataset primarily focuses on morphological and histological outcomes, we agree that assessing downstream signaling components would provide important mechanistic context. We have therefore added a discussion paragraph acknowledging this limitation and outlining plans for future studies that will examine Epac1/2, PKC-ζ, and MAPK phosphorylation as potential mediators of Rap1a-RAGE–dependent remodeling.

  1. Sample Size and Statistical Power

Reviewer Comment:
The study uses a relatively small sample size (n = 5–6 per group). The authors should include a power calculation or discussion to justify that the study was adequately powered to detect the observed effects. Some statistical comparisons (e.g., VW/BW ratios, collagen quantification) approach significance, suggesting that marginal increases in sample size could yield more definitive conclusions.

Response:
We thank the reviewer for this thoughtful point. A post-hoc power analysis has been performed, and we have added a brief description of this analysis in the Methods section and discussed the implications on the limitations in the Discussion.

  1. Sex-Based Limitation

Reviewer Comment:
All experiments were conducted using male mice. Given known sex differences in cardiovascular remodeling and RAAS activity, this limitation should be clearly stated and discussed. The inclusion of female cohorts in future studies would enhance translational relevance.

Response:
We fully agree with this comment. We have now explicitly stated in the Methods that only male mice were used and added a detailed paragraph to the Discussion (acknowledging this limitation.

  1. Functional Correlation

Reviewer Comment:
Echocardiographic findings demonstrate structural remodeling with preserved systolic function. However, discussion of the physiological implications (e.g., diastolic dysfunction, potential HFpEF phenotype) would add depth and better link the findings to clinical hypertensive heart disease.

Response:
We appreciate this excellent suggestion. We have expanded the Discussion to include a section on functional interpretation.

  1. Figure and Data Clarity

Reviewer Comment:
Although figures are generally well presented, some could benefit from improved clarity: (a) Include scale bars and magnifications in all histological figures; (b) Use consistent labeling for groups (WT, RAGE KO, RapKO) across all panels; (c) Clarify whether error bars represent SEM or SD in figure legends.

Response:
We have carefully revised all figures as follows:

  • Scale bars and magnifications have been added to all histological panels though they may not be visible due to size constraints. We will include larger images in Supplement 2.
  • Group abbreviations have been standardized throughout to WT, RAGE-KO, RapKO, and this convention is used consistently across all figures, legends, and text.
  • Figure legends now clearly specify that error bars represent the standard error of the mean (SEM).

Minor Comments

  1. Typographical and formatting inconsistencies
    We have performed a comprehensive proofread to correct spacing, punctuation, and abbreviation inconsistencies throughout the manuscript, and hopefully have corrected them all.
  2. Condensing RAGE–AT₁R interaction description
    The introductory section on RAGE–AT₁R crosstalk has been condensed for clarity.
  3. References for Epac1/Epac2 dual roles
    We have added two citations supporting dual Epac1/Epac2 activity in hypertrophic signaling.
  4. Abbreviation definitions
    All abbreviations (e.g., NHE3, CO) are now defined at first mention. A consolidated Abbreviation List has been added to the end of the manuscript for reader convenience.
  5. Pharmacological modulators of Rap1a/Epac1
    We agree this is an important translational consideration. A short paragraph has been added to the Discussion referencing emerging Rap1a activators (e.g., 8-pCPT-2′-O-Me-cAMP, ESI-09 inhibitors) and RAGE antagonists (FPS-ZM1) as potential therapeutic leads to modulate this signaling axis in hypertensive remodeling.

We are grateful to the reviewer for their constructive feedback and encouraging assessment of our study. These thoughtful suggestions have substantially improved the manuscript’s scientific depth and presentation. We hope the revisions satisfactorily address all concerns and that the revised version meets the journal’s standards for publication.

Reviewer 3 Report

Comments and Suggestions for Authors

This manuscript is a scientifically well-founded and well-structured study that investigates the molecular mechanisms underlying hypertension-induced cardiac remodeling, with particular emphasis on the roles of the RAGE and Rap1a signaling pathways. The research is novel, as it experimentally examines the functional interaction between these two pathways and the angiotensin II–induced alterations using knockout mouse models. The methodology is detailed, and the experimental protocol is carefully documented, ensuring the reliability and reproducibility of the findings. To further strengthen the manuscript, it would be advisable to:
– briefly summarize the translational relevance of the animal study results to human models
– clarify in the discussion whether the observed AngII effects are primarily linked to hemodynamic overload or to cell-level signaling pathway activation
– consider further investigation of the direct molecular interaction between Rap1a and RAGE (eg. through protein–protein interaction studies or phosphorylation status analyses)
– briefly address the potential influence of sex-related differences on the experimental outcomes, as the study was conducted in male mice, while RAGE and Rap1a signaling may be modulated by hormonal factors
– mention whether any pharmacological approaches are known to induce similar protective effects as those observed in the genetic knockout models, as this would enhance the translational and clinical relevance of the study

It would also be useful to ensure that the figures are arranged in the order of their first appearance in the text., Overall this is a high quality , scientifically rigorous and relevant study that sheds new light on the role of Rap1a and RAGE in the pathomechanisms of hypertension and provides a valuable contribution to the understanding of molecular targets in cardiovascular remodeling.

Author Response

Response to Reviewer #3

Manuscript Title: Knocking Out Rap1a Attenuates Cardiac Remodeling and Fibrosis in a Male Murine Model of Angiotensin II–Induced Hypertension
Authors: Cody S. Porter, Larissa T. Brown, Can’Torrius Lacey, Mason T. Hickel, and James A. Stewart, Jr.
Journal: Cells (2025, Volume 14)

We sincerely thank the reviewer for their positive and encouraging assessment of our work. We greatly appreciate the recognition of our study’s novelty, methodological rigor, and contribution to understanding the interplay between RAGE and Rap1a signaling in hypertension-induced cardiac remodeling. We have carefully revised the manuscript in accordance with the reviewer’s constructive suggestions, as detailed below.

Reviewer Comment 1: Translational relevance of animal findings

Comment:
Briefly summarize the translational relevance of the animal study results to human models.

Response:
We appreciate this suggestion and have expanded the Discussion section to emphasize how our findings may translate to human cardiovascular pathology. Specifically, we now discuss the clinical parallels between AngII-induced remodeling in mice and hypertensive heart disease in humans, highlighting that RAGE and Rap1a are upregulated in human myocardium during hypertensive stress and fibrosis. We also note that targeting Rap1a signaling could provide a mechanistically distinct approach to mitigating adverse remodeling independent of systemic blood pressure modulation.

Reviewer Comment 2: Mechanistic origin of AngII effects

Comment:
Clarify in the discussion whether the observed AngII effects are primarily linked to hemodynamic overload or to cell-level signaling pathway activation.

Response:
We thank the reviewer for this insightful point. We have revised the Discussion to clarify that although AngII infusion increases systemic blood pressure, our data did not include hemodynamic load and was added as a limitation to the discussion.

Reviewer Comment 3: Direct molecular interaction between Rap1a and RAGE

Comment:
Consider further investigation of the direct molecular interaction between Rap1a and RAGE (e.g., through protein–protein interaction studies or phosphorylation status analyses).

Response:
We fully agree with this recommendation. While such analyses are beyond the current study’s scope, we have added a paragraph to the Discussion  describing ongoing and planned experiments to evaluate Rap1a-RAGE interaction through co-immunoprecipitation and phosphorylation assays. We cite relevant literature demonstrating Rap1a’s involvement in cytoskeletal and receptor-linked signaling and suggest that future work will determine whether RAGE-mediated Rap1a activation contributes to AngII-induced remodeling via specific downstream targets such as Epac1 and PKC-ζ.

Reviewer Comment 4: Sex-related influences

Comment:
Briefly address the potential influence of sex-related differences on the experimental outcomes, as the study was conducted in male mice, while RAGE and Rap1a signaling may be modulated by hormonal factors.

Response:
We appreciate this important point. We have added text to the Methods explicitly stating that only male mice were used in the present study, and a corresponding statement in the Discussion  acknowledging this as a limitation.

Reviewer Comment 5: Pharmacological modulation and translational relevance

Comment:
Mention whether any pharmacological approaches are known to induce similar protective effects as those observed in the genetic knockout models, as this would enhance translational and clinical relevance.

Response:
We thank the reviewer for this suggestion and have included a short discussion of pharmacological modulators in the Discussion. We reference RAGE antagonists (FPS-ZM1, azeliragon) and Epac1 inhibitors (ESI-09), as well as Rap1a activators (8-pCPT-2′-O-Me-cAMP), noting that these agents have shown promise in reducing cardiac or vascular fibrosis in preclinical studies. This addition underscores the potential for therapeutic translation of our genetic findings to pharmacological interventions in human hypertension and heart failure.

Reviewer Comment 6: Figure order and consistency

Comment:
Ensure that the figures are arranged in the order of their first appearance in the text.

Response:
We have reviewed the entire manuscript to confirm that all figures now appear sequentially according to their first mention in the text. Figure numbering and citations have been updated accordingly for clarity and consistency.

We are grateful for the reviewer’s highly positive and thoughtful evaluation of our study. The suggested refinements have helped improve the manuscript’s clarity, translational focus, and scientific depth. We believe the revised version more effectively communicates the relevance of Rap1a and RAGE signaling to hypertensive cardiac remodeling and its potential implications for future therapeutic development.

Round 2

Reviewer 1 Report

Comments and Suggestions for Authors

The revised manuscript has been improved, no further recommendation.